# Effect of Extrusion Parameters on Short Fiber Alignment in Fused Filament Fabrication

**DOI:** 10.3390/polym13152443

**Published:** 2021-07-24

**Authors:** Patrick Consul, Kai-Uwe Beuerlein, Genc Luzha, Klaus Drechsler

**Affiliations:** Chair of Carbon Composites, Department of Aerospace and Geodesy, Technical University of Munich, 85748 Garching, Germany; kai.beuerlein@tum.de (K.-U.B.); genc.luzha@tum.de (G.L.); klaus.drechsler@tum.de (K.D.)

**Keywords:** fused filament fabrication, additive manufacturing, composites, short carbon fibers

## Abstract

Additive manufacturing by material extrusion such as the widespread fused filament fabrication is able to improve 3D printed part performance by using short fiber reinforced composite materials. Fiber alignment is critical for the exploitation of their reinforcing effect. This work investigates the influence extrusion parameters have on the fiber alignment by conducting set of experiments on the process parameters determining whether the flow under the nozzle is convergent or divergent. A strong impact of flow conditions during extrusion line shaping on the fiber alignment is observed and two extremes are tested which show a large difference in strength, stiffness and strain at break in tensile testing along the extrusion lines. From highest to lowest fiber alignment, strength is reduced by 41% and stiffness by 54%. Fiber misalignment also leads to inhomogeneous strain fields in the layers when tested perpendicular to the extrusion lines. It is demonstrated that material flow after the nozzle has a high impact on the material properties of short fiber reinforced 3D printed parts and needs to be considered in process design.

## 1. Introduction

Additive manufacturing (AM) by material extrusion, such as the widespread fused filament fabrication (FFF), produces parts directly from digital files by melting plastics in a hot nozzle and depositing it along a toolpath. This toolpath is arranged in layers which are stacked on top of each other to form the part. Through this line-by-line and layer-by-layer approach the process is able to generate complex parts and leverage the reported advantages of AM such as:
freedom of design and complexity [1,2,3,4,5,6];tool and mold free production with reduced lead times [3,4,7,8];part consolidation by printed assemblies [2,4,6,9]economies of scope and low cost for parts in small production volumes or individual production [1,3,4,5,7,8,10,11]smart materials & functionalization [1,3,4,6,9];high efficiency with low material waste and lightweight designs [1,3,4].

However, the same principle also causes some of the challenges AM faces:
anisotropic material properties [2,11,12,13,14,15,16];surface quality and accuracy [3,4,6,9,16,17];low production speed [1,4,9,15];
caused by the deposition of material along the toolpath in the shape of extrusion lines. The rounded sides of the extrusion lines cause a wavy surface, the interfaces between them can form weak bonds, the continuous melting of material in the printhead and the kinematic movement limit the production speed.

Several articles have proposed ways to overcome the challenge of anisotropic properties yet anisotropy in the part can have different causes. To explain more in detail, we introduce a local coordinate system in the material, based on the toolpath, to describe the material properties in Figure 1 that will be used in this article.

We define:X—along the toolpath;Y—in the layer plane perpendicular to the toolpath;Z—perpendicular to the toolpath and the layer, in most cases this is also the tool axis.

The most often investigated anisotropy is due to lower properties along the Z axis than in plane. This is caused by incomplete bonding of the layers to each other and is usually described by the fusion bonding model [18] on the welding processes of the plastic between layers focusing on aspects such as pressure driven flow [19], polymer crystallization rate [20] or DoE’s looking at the parameters controlling the temperature of the part [21,22]. 

Y anisotropy is caused either by macro-pores running along the edges of the extrusion lines limiting the contact [23], incomplete welding of the lines to each other [22] or through the addition of reinforcing fibers which align along X [24]. Anisotropy from the orientation of the lines in a part is essentially a Y anisotropy with changing orientations of X and Y as we use a toolpath-based coordinate system.

Fillers and fibers are often added to polymers to improve their properties, either their stiffness and strength but also other properties like thermal stability [25,26]. While the impact of near spherical fillers such as glass beads or powders depends generally only on their content in the polymer, fibers have an anisotropic effect. The contribution of a fiber is greater for longer fiber lengths with the highest performance offered by continuous fibers. They greatly improve the properties of the material along the fiber direction while the transverse effect is very small. Taking advantage of the anisotropic properties provided by the reinforcement and their low density allows for lightweight design of high performing structures that, thanks to the processability of the matrix material, can be scaled to large sizes. This has long established composite materials in aerospace applications [27,28] and because of their ease of processing combined with exceptional properties and fatigue resistance they have since expanded into countless other applications, including many not typically associated with lightweight design such as construction for structural repair [29] or railway sleepers [30]. For many new applications, cost is no longer second to material performance as in aerospace, and the materials are increasingly used to take advantage of the production processes, ease of handling and the possibility to integrate functionality through the material composition [31]. 

In short-fiber composites the contribution of the reinforcements is limited by the fiber length but also their degree of alignment [32]. Like continuous reinforcements, short fibers improve the material properties along their orientation. However, as stresses cannot be transmitted through the fiber over the component, their contribution to strength and stiffness is limited by the fiber length, or more accurately the surface area over which stresses can be transmitted into the matrix [33]. Not just fiber length and content [34], but also their surface and its compatibility with the matrix have significant impacts on the mechanical properties [35]. Compared to continuous fibers, the orientation of short fibers is much more difficult to control and is strongly influenced by processing conditions as the fibers are free to rotate with the flow. In injection molding there is significant effort put into determining the flow fields inside molds and flow channels as the changing thickness of material results in different flow fields and fiber orientations [36,37,38,39,40]. 

AM by material extrusion can greatly simplify the prediction of short fiber orientation in complex parts with changing wall thicknesses as the part is composed of extrusion lines repeated in Y to the desired wall thickness and stacked in Z to form the parts. Once the material is deposited the matrix solidifies and the fiber orientation is fixed. Research on predicting the fiber orientation in extrusion based AM thus far has been focused first on the flow inside the nozzle and the die swell when extruding the material into air, rather than onto a platform with relative movement [41,42,43,44]. In a second step, this research has been extended to planar flow, considering deposition onto a moving platform but flow only in the central plane of the nozzle and extrusion line with the fiber orientation changing from the tool direction to the movement direction along the flow lines [45,46]. This This assumes that the extrusion line has the same width as the nozzle and the change in flow direction can be simplified to two dimensions. A high orientation after the change in flow direction was found and has been used in other publications to functionalize parts by taking advantage of the high degree of alignment by adapting the toolpath [47,48]. The fiber alignment in one of these investigations was verified by micro-CT images and was found to decrease when the material is deposited onto a moving platform compared to being extruded into air [48]. The influence of different angles of the internal geometry of the nozzle in the convergent section before the orifice on the pressure necessary for extrusion has also been investigate in terms of fiber shortening [49]. In direct write AM a nozzle with changing orifice diameter has been reported that can influence the fiber orientation by changing from a convergent to a divergent flow [50]. 

A divergent flow can also be achieved in extrusion-based AM by FFF after the nozzle by extruding the material to a line width larger than the nozzle diameter and forcing the material to change flow direction not just from Z to X but also to Y and rather than a 2D flow state to a 3D flow. 

To our knowledge this has not been investigated yet and most published investigations are simulations of the flow. To address this identified research gap, we pose the research question whether the extrusion of lines wider than the nozzle diameter cause a flow divergence to misalign fibers along Y rather than purely along X that is strong enough to significantly affect the mechanical properties of a printed part with multiple extrusion lines and layers.

## 2. Materials and Methods

### 2.1. Materials

Initial screening experiments with filaments of different manufacturers, with different fiber contents and matrix materials showed that all possess similar properties regarding fiber orientation and distribution. This initial screening compared Nylon and PETG as matrix materials and fiber contents of 10, 15 and 20%. All showed similar behavior so for the experiments reported in this article a 20% carbon fiber reinforced PETG filament by “Formfutura” named “CarbonFil” was used. High fiber content was chosen to maximize the effect of changes in fiber alignment on material properties and amorphous matrix was chosen to minimize crossover effects from matrix welding. A high fiber content was expected to give a more accurate result when measuring fiber misalignment as more fibers would be visible in the cross-sections. The “CarbonFil” material was dried for five hours at 65 °C before printing of the specimens and kept in a box with a moisture absorber during printing. According to the manufacturers data sheet the material has a density of 1.19 g/cm^3^, water absorption of 0.13%, tensile strength of 52.5 MPa, tensile modulus of 3800 MPa and Vicat softening temperature of 85 °C at 0.455 MPa. Print temperatures are recommended at 230 to 265 °C for the extruder and up to 85 °C for the print bed.

Specimens were printed on a modified delta printer based on a Tevo Little Monster running on a Duet2 control board with an E3D Volcano hot end with 60 W heating power and a type K thermocouple for temperature control using stainless steel nozzles. The build chamber was enclosed but not heated, the print bed was heated with a 400 W silicone heater controlled by a 100 kΩ thermistor underneath a glass plate. Temperatures were set at 265 °C for the extruder and 75 °C for the print bed, within the recommended range. Extruder temperature was chosen as high as possible to minimize the viscosity and allow maximum material flow. The bed temperature chosen was high but below the Vicat softening temperature to ensure bed adhesion and allow the material to solidify and stabilize the extrusion lines. 

The fiber lengths and orientations in the filament before extrusions can be seen in Figure 2, based on measurements at five positions and a total of 315 recognized fibers of minimum aspect ratio of 3:1.

### 2.2. Method

The aim was to investigate how to control fiber alignment through extrusion parameters, by creating a flow divergence under the nozzle. For this we started by choosing relevant process parameters which were expected to affect the fiber alignment and set up a design of experiments. For these experiments we printed a first set of specimens to analyze the fiber alignment and extrusion line shape without interactions with neighboring extrusion lines to identify the parameter sets with highest and lowest fiber alignment. Using the identified parameter sets of highest and lowest fiber alignment, we finally printed a second set of specimens for tensile testing. Testing along the extrusion lines in X and perpendicular to them along Y where the misaligned fibers were oriented. During tensile testing we used digital image correlation (DIC) to observe the strain fields and determine the impact of the changed fiber alignment on the mechanical properties both over the specimen and at an extrusion line level. 

The following sections will explain in detail which parameters were chosen and why they were selected, the design of experiments, testing procedure and equipment used.

#### 2.2.1. Parameter Identification & Design of Experiments

To identify the extrusion parameters, which may influence fiber alignment, we started with a general consideration of the effect shear flow has on the fibers. A convergent flow or stretching of the material will align the fibers in the flow or stretching direction. A divergent flow will orient them perpendicular to the main flow direction in the divergence direction. An advancing front on a cold wall will have a highly stretched skin layer at the cold wall [51].

For the extrusion process, we identified four phases and their relevant parameters influencing fiber orientation:Within the nozzle there is convergent flow in a narrow channel with flow along the tool axis Z due to the inner nozzle geometry. Nozzle diameter, polymer viscosity and flow velocity are the main factors.Under the nozzle, the flow turns to follow the toolpath. The layer height is usually lower than the nozzle diameter, therefore the flow is convergent in X with a cold wall on the part, that should result in the aforementioned stretched layer. Layer height in relation to the nozzle diameter is added to the list of relevant parameters to describe the degree of convergence of the gap between nozzle and part into which the material is pushed.The extrusion line may be wider than the nozzle and therefore become divergent in Y. This occurs in parallel to phase two and only if the extrusion line width is larger than the nozzle diameter, making the material flow to the side rather than just planar as in [45,46]. The relevant parameter is the extrusion line width in relation to the nozzle diameter, again a relative factor is used to describe the degree of divergence, as a pure line width would carry no information on the divergence of flow. A 0.6 mm line printed by a 0.4 mm nozzle has a divergence, while the same line printed by a 0.6 mm nozzle is purely planar flow.As the nozzle moves over the deposited material the material can be stretched, as the bottom of the material is standing still, while the nozzle edge moves over it in contact with the material. The parameters are the layer height and the speed of the nozzle.

The parameters movement speed and flow velocity are coupled through the line width, layer height and nozzle parameters. To keep the focus on the flow in and around the nozzle flow, velocity was chosen for the DoE. Flow velocity in this article refers to the average velocity at the nozzle outlet and is determined by dividing the flow rate of the extrusion line by the nozzle orifice area. Print speed is calculated from the flow rate and the extrusion line cross-section. Using the flow velocity or flow rate to determine print speed should ensure that the extruder is able to maintain the flow when larger extrusion lines are used and make the flow in phases two and three more comparable between experiments.

Polymer viscosity is affected by the material, its temperature and the shear rate which makes it very difficult to control in a dynamic process like AM. In addition, extrusion temperature is an important control parameter for the Y and Z anisotropy and layer bonding [20,21,22], generally a dominant issue when printing parts. The viscosity is therefore removed from the set of control parameters, as we cannot accurately control it and changes in temperature may be unfeasible for real applications. Layer height and nozzle geometry where also found to affect the fiber orientation in [45]. However, as they considered the flow to be planar, extrusion width did not play a role and over-extrusion was considered not by a widening line but by a height increase after the nozzle and a bullnose flow profile resulting in lower fiber orientation. In the current work the aim was to use a flow divergence after the nozzle to reorient the fiber along Y.

This results in the parameters of Table 1, their squares and interactions:

Using the relation of extrusion width and layer height to the nozzle diameter, a better impression of the nature of the flow in phases two and three is given. An extrusion width of 100% of the nozzle diameter skips phase three completely resulting in a flow similar to the planar flow described in [45,46], while a 200% extrusion width has a strongly divergent flow. Layer height was chosen by experience, based on previous experiments, where values larger than 60% of the nozzle resulted in poor contact with the previous layer. The lower limit was dictated by the printers step resolution of 0.08 mm and smallest nozzle of 0.4 mm, giving a minimum layer height to nozzle diameter of 20%.

The effect of interactions was determined by the DoE as well. Due to the relative nature of the parameters for extrusion height and width their interactions with the nozzle diameter are the absolute width and height: *w* × *D* is the extrusion line width in mm and *h* × *D* is the layer height in mm.

The parameters were input into Sartorius’ Modde 13 to generate a determinant optimal (D-optimal) DoE. Using 27 design runs and three center points in a quadratic model, the design has a power of 95; 15.67 I-optimality and orthogonality condition number of 5.91 with a total of 30 experiments. A full factorial design would have similar numbers, at power 100, I-optimality 16.7 and orthogonality condition number 5.39 at a much higher number of 84 experiments. D-Optimal designs are computer generated designs which maximize the determinant of the information matrix. This means the experiments span the experimental region as well as possible by maximizing their volume in the experimental space. The design consisted of 27 design runs and three repeated runs at the center point to judge response repeatability of the screening. The parameter levels of the experiments and results are all listed in Appendix A Table A1. The print parameter settings in the slicer Simplify3D which were not changed between the experiments but determine the toolpath generation are listed in Appendix A Table A2. 

The observed responses for the DoE are:Fiber alignment in the XY plane, the aim information of this articleActual extrusion width/setpoint extrusion width, this is to ensure the extrusion process achieves the desired line geometry and is stable, without filament slipping.Surface quality on the XZ side of the printed specimens, this is also to ensure process stability. An irregular wall indicates an unstable extrusion process.

Fiber alignment is the target response that the experiments were designed for. The second and third responses are for quality control of the process to ensure the setpoint is achieved and the extrusion process is stable. As the surface roughness of printed parts also depends largely on the layer heights, information on the surface roughness is practically a byproduct of the experiments and reported as well but will not be analyze in the same depth. 

The experiments to are done by printing one cube of 35 mm sides of a single extrusion line wall for each of the 30 parameter sets. All parameter sets of the same nozzle diameter were nested and printed at the same time. This ensures the previous layer has enough time to cool down and is not deformed by the extrusion of the next layer. To identify the cubes after printing, each was printed with the experiment number embossed in one of the sides by 0.1 mm which was visible after printing. They were printed as single walls to avoid interactions between the extrusion lines within a layer.

After the experiments the parameter sets with the highest and lowest fiber orientation and a stable extrusion process were chosen to print tensile test specimens for testing along X and Y using digital image correlation (DIC) to observe the strain. All cubes and tensile test specimens were printed from the same roll of material to avoid differences between batches affecting the results. This was to investigate the effect the fiber misalignment can have on the mechanical properties, over tensile specimens as a whole but also at the level of individual extrusion lines using the strain field to visualize the extrusion line interfaces when testing along Y.

#### 2.2.2. Analysis

After printing of the cubes, the first analysis step was surface scanning using a Keyence VR500. This 3D scanner with a measurement accuracy in of height profiles of ±2.5 µm was used for the extrusion line shapes and wall surface roughness. Width accuracy was ±2 µm, used for measuring the actual line width. The extrusion line was scanned in the XY plane and the walls of the cubes in XZ plane. This allowed us to measure the extrusion line width, height profile and shape and their effect on the surface roughness of the walls, which is affected by the layer height, but also artefacts which may form due to unstable extrusion. Using automated batch analysis and edge detection of the scans to orient the specimens in the measurement it was ensured that the surface roughness measurements in XZ were always performed on the same area in the same central region of the wall, leaving space to the corners, as shown in Figure 3. These measurements were performed to ensure process stability to aid when selecting the parameter sets of highest and lowest fiber misalignment, by ensuring the processes were also stable and repeatable.

After the scans, one of the walls was cut from the cube and embedded in epoxy resin. This was ground and polished to obtain cross-sections of the XY plane for fiber orientation as well as the XZ and XY planes to confirm the fiber orientation was indeed mainly in the XY plane as expected and reported in literature [48]. This was the case as micrographs showed fiber alignment in XY, so XZ images showed fibers only in X. Circular cross-sections of the fibers were only visible where the fibers pointed in Y. YZ images showed fibers only in specimens where a high degree of misalignment was found in the XY plane, otherwise only circular sections of the fibers were visible.

To measure the fiber misalignment, the orientation of the individual fibers within the extrusion line was measured. For this, microscope images were taken of the polished cross-section using 10× magnification and multiple image alignment on an Olympus BX41M to show the entire width of the extrusion line. A Python 3.9 program was used to measure the orientation of the individual fibers, that:Transformed the image colors into a grayscale.Applied a threshold to transform the grayscale images into a binary of white fibers and black surroundings.Identified objects of white pixels and applied a threshold to delete smaller objects to filter and remove reflections of scratches or particles. In this case a threshold of 21 µm^2^ was used.Calculated the object location to then calculate the length of the main axis, the width perpendicular to it and orientation.It then classified the objects into fibers and large particles by considering objects with a length to width ratio of 3:1 a fiber and everything else a particle.Finally, it output the length and orientation distribution of the three classes, objects, fibers and particles to create plots of different statistical visualizations, such as histograms and cumulative curves as well as saving the data of the objects.

As the histograms of the individual fiber orientations from the X axis proved to be normal distributions around the X axis, the misalignment of fibers can be described using the standard deviation of the distribution. To consider local variations this was repeated at three different positions of the cube and the results were averaged for the fiber misalignment.

For tensile testing, two extremes of the parameter sets were chosen, one with a minimum fiber misalignment and one with maximum misalignment. Three specimens were printed of each parameter set and for each testing direction X and in Y, corresponding to a 0° and 90° infill or fiber orientation. Specimens were printed without perimeters around the outside using only parallel lines to avoid having 0° perimeters around 90° specimens influencing the mechanical properties. Dimensions were 150 mm × 24 mm × 2.4 mm in X × Y × Z. Width and height were chosen to allow for a whole number of lines and layers to be printed for both parameter sets. The specimens were smoothed with a grain 240 sandpaper. This total of twelve specimens was airbrushed on the surface with a stochastic black and white pattern that could be observed using DIC. The universal test machine was a 100 kN UPM by Hegewald & Peschke using a 10 kN load cell. DIC was performed using a GOM ATOS Capsule in Aramis mode. For the stress–strain curves a virtual extensometer was calculated in DIC over a length of 25 mm. The strain field was also calculated and used to check for local differences in strain of the specimens which were found only for the specimens tested along Y, corresponding to a 90° infill/fiber orientation.

## 3. Results

In this section examples of the results for the 30 experiments of the DoE are presented and explained, going into detail in the following subsections. Appendix A Table A1 shows the factors of the 30 experiments and their results. Experiment N3 did not produce a usable cube and had to be excluded. It is noteworthy that N4 and N5 have the same material velocity and nozzle size as N3 yet were able to print cubes. It appears that the print of cube N3 did not fail due to under extrusion and filament slipping, but rather that at the high movement speed, the material was torn of the previous layer by the nozzle during phase four of the previously mentioned phases. The column “Extrusion Line Shape” of Appendix A Table A1 refers to the height profiles of the XY scans and will be explained in Section 3.1.1. “Extrusion Line Shape and Width”. 

### 3.1. Extrusion Shape and Surface Quality by 3D Scans

#### 3.1.1. Extrusion Line Shape and Width

Scans of the XY plane on the topmost layer of the cubes give feedback about the geometry of the extrusion line that allow conclusions about the stability of the extrusion process. The measurements using the Keyence VR5000 scanner provided 3D images which measure the width of the extrusion line and determine the height profile and shape of the extrusion line. Height profiles were determined perpendicular to the extrusion line edge, on ten lines 0.75 mm apart from each other that were then averaged. Four qualitative kinds of extrusion line shapes can be identified which were named “convex”, “flat”, “concave” and “jagged” based on the shape of the height profile seen in Figure 4.

A tendency was visible as large layer-to-diameter ratios of 60% lead to convex shapes, between 40% and 50% the shape was flat and for smaller layers the extrusion line cross sections became concave. Low layer height-to-diameter ratios and high speeds lead to jagged extrusion line edges that indicate an instable printing process, but not extrusion process. The extrusion was torn off the part by the nozzle as it passed the freshly extruded material at high speed without the material having time to bond to the substrate. Wider lines seemed to counteract this effect, possibly by increasing the contact pressure between the extrusion line and the substrate [52]. Convex shapes tended to be slightly wider than their setpoint while concave shapes were slightly narrower. The large layer heights probably began flat, but as the material retains heat for a while, this caused the edges to sag downwards giving the convex shape. Another possibility is that due to high extrusion pressure a swelling effect caused the center to expand upwards as the nozzle passed on, similar to a die swell effect [45]. Concave layers probably cooled down quickly from outside to inside, warping slightly, pushing the edges up. Flat layers in between appeared to keep a balance between the two, indicating that, by extrusion line shape, a layer height to nozzle diameter ratio of 40 to 50% is ideal.

In these images the edges were easily visible and could be used to determine the extrusion line width. Using edge detection, a straight line was fitted to both sides and the distance between them was measured at three evenly spaced positions about 1 mm apart, which is then averaged, though the typical difference between the measurements was about 0.002 mm, which is the measurement accuracy of the scanner in the XY plane, indicating excellent parallelism of the edges. Normalized by the set extrusion line width, the values of column three of the measurements in Appendix A Table A1 were calculated to indicate extrusion stability. This would be used to help select a parameter set for the tensile test specimens as a quality control parameter to remove parameter sets which lead to unstable extrusion. Extrusion lines are deemed stable if their width is more than 95% of their respective setpoint and they did not have jagged edges. As the only extrusion lines with significantly more than 100% setpoint width were jagged lines, no upper limit was set.

#### 3.1.2. Surface Roughness

Surface roughness was determined by 3D scans as area roughness *Sa* by scanning a square of the wall of the cubes as shown previously in Figure 3. This means that roughness is influenced by the layer lines which are easily visible in the example, this is in fact rather a surface waviness, rather than roughness. However, surface artefacts that are caused by jagged lines also appear in a smaller form on concave extrusion lines can significantly increase the surface roughness. This is a true roughness, as the artefacts result from the sharp, jagged edges of the extrusion lines. Figure 5 shows three examples of the surfaces, with color-coded height profiles.
N11 with the lowest surface roughness, due to stable extrusion in flat extrusion lines and low layer height of 0.16 mm resulting in a homogeneous surface waviness.N14 with very stable but convex extrusion lines in large layers of 0.48 mm, showing a still homogeneous but larger wavinessN23 with the roughest surface, even though the layer height of 0.12 mm is the lowest of the three examples, due to surface artefacts from the jagged extrusion line shape.

Nozzle speed coupled with the layer height can influence the surface roughness quite drastically by creating these surface artefacts of jagged extrusion edges. However, when optimizing a printing process, an unstable extrusion resulting in jagged lines is unacceptable. If the experiments with jagged lines are eliminated from the DoE model, it is clear that surface waviness is dominated by the layer height as the five largest effects are the nozzle diameter, the layer height ratio and its interactions. The effect of the flow velocity in the nozzle is small, yet cannot be eliminated from the model, as seen in Figure 6.

A 4D contour plot of the model is shown in Figure 7. It can be seen that low layer heights and nozzle diameters reduce the surface roughness measurement results, while the highest surface roughness, or waviness, is expected for large layer heights. The surface roughness seems to increase with extrusion line width yet flow velocities improve the surface on large layers.

### 3.2. Fiber Alignment in XY

The main result of this work is the effect of extrusion parameters on fiber alignment which are presented in this section. The measurement is based on microscope images that are first transformed to binary images in which the fibers were detected, and their individual orientation was measured and color-coded. Finally, a histogram of the fiber orientation was output and the standard deviation of the normal distribution was determined. The steps are shown in Figure 8 for two extreme cases, N19 and N14, later used in tensile testing:

The resulting standard deviations of the fiber orientation histograms were used to judge fiber alignment and calculate a model for the effects of the parameters in the DoE software, shown in Figure 9. As expected, the strongest effect was through the extrusion width/nozzle diameter, followed by the interaction with layer height/nozzle diameter; the two parameters which describe the flow convergence or divergence under the nozzle. It also shows that the effect of flow velocity in the nozzle was smallest among the considered effects. The largest effect caused by the flow velocity was the interaction with the layer height/nozzle diameter, the parameter describing the stretching of the extrusion line by the moving nozzle, previously described as phase four.

Compared to the effect on surface roughness, the effects were much clearer for the fiber orientation, as the confidence interval span was smaller compared to effect magnitude. The effects are again plotted in a 4D contour in Figure 10 to better visualize the interaction between the parameters and find ideal settings for minimizing or maximizing the fiber misalignment:

With increasing nozzle size, a higher range of misalignment becomes possible, probably as the length of the out-of-plane flow increases. With higher flow velocity, a higher degree of alignment is achievable. A setpoint of extrusion width and layer height can be identified for each nozzle diameter for which changes in velocity have no impact on the fiber orientation. This is important because it allows the determination of fiber alignment using the two parameters and frees up the velocity and nozzle diameter as control parameters to control the time in which a layer is completed when trying to improve the layer adhesion.

### 3.3. Tensile Tests on High and Low Fiber Aligned Specimens

To investigate the effect different degrees of fiber alignment have on the mechanical properties of 3D printed parts, tensile test specimens were printed. Based on the results in Appendix A Table A1, experiments N14 and N19 were chosen. Experiments with large nozzle sizes were chosen as they offer the largest achievable difference in fiber alignment compared to smaller nozzle diameters. Additionally, the larger extrusion lines can show potential local differences in strain fields better during DIC as the resulting extrusion lines are wider. N15 had an even higher fiber misalignment than N19 but showed almost 12% under-extrusion, compared to 3% under-extrusion in N19. N19 was chosen over N22 respecting a higher degree of misalignment while accepting the negligible amount of under extrusion, based on the quality assurance criterion defined at a minimum of 95% actual extrusion line width of the setpoint width.

Figure 11 shows the stress–strain curves of the resulting experiments. The experiments of the same parameter set show little variance between them, but the difference between those and the other parameter sets was significant. While the specimens of N14 with low fiber misalignment had high stiffness and strength when tested along the extrusion lines, the N19 specimens with high fiber misalignment have larger strain at break. Surprisingly the difference in strength and stiffness of the specimens tested along Y is much lower, however the N19 specimens still achieved higher strain at break.

Table 2 and Figure 12 show the averaged values of the experiments and calculated values for the elastic modulus and toughness of the specimens. Tested along X, the N14 specimens with an average standard deviation of fiber alignment of 9.13° have 57% of the strain at break, 170% of the strength, and 217% of the stiffness of the N19 specimens with an average standard deviation of fiber alignment of 30.43°. 

Strength and stiffness of the highly aligned specimens were higher than the data sheet values from the filament manufacturer, with 52.5 MPa tensile strength and 3800 MPa tensile modulus, indicating that for the data sheet specimens were printed with infill at an angle to the test direction.

The toughness of N14 is 90% that of N19 with high variations as both parameter sets had an outlier with exceptional strength at break, if the outliers are excluded, the relation stays at 90% N14/N19 at a reduced standard deviation of 9.07 and 6.56 J/mm^3^, respectively, and average of 823.52 and 885.37 J/mm^3^.

Along the Y direction, the differences were significantly lower and strength and stiffness were almost identical. Considering that even for the highly misaligned fibers the standard deviation was only 30.43°, this means that 68% of the fibers are arranged within the ±30° region and 95% within ±61°. From the tested Y axis perspective, only 32% of the fibers are at less than 60° and only 5% at an angle smaller than 29°. Fibers oriented at more than 45° from the tensile stress direction have almost no effect on the strength and even at 29° their contribution to the strength is a fraction of their contribution when aligned with the stress [53] (p. 117). Additionally, with high fiber misalignment in the core of the extrusion line, the edges of the extrusion line still contain fibers mostly aligned along X, as displayed in Figure 13. Fibers along X are colored green, the fibers along Y are red. This results in a weak spot at the interface between the extrusion lines, further weakened by bonding issues or pores between the lines, explaining the similar mechanical properties.

This is visible in the strain field of the specimens tested along Y as well. With the extrusion lines still visible at the edges, it is clear that a region of high strain exists between the lines, visualized in red, and the extrusion line center is less strained, which indicates it is stiffer. Looking at the strain field of N19 tested along Y in Figure 14, at a strain of 1% averaged along 25 mm, differences of more than 0.5% strain, 50% of the average along the center line, occur periodically at the distance of one extrusion width.

Meanwhile the strain field of sample N14 with a high degree of fiber alignment showed a much more homogeneous strain field at an average strain of 1% in Figure 15. Some spots of higher strain are still visible, however much lower in number and magnitude. These strain differences may be caused by porosity between the extrusion lines.

Following up on the differences in local strain, a local elastic modulus along the centerline was calculated from the strain fields at 5 MPa and 10 MPa. This stress is in the elastic region for both specimens. The moduli curves are visualized in Figure 16. The curves spike significantly. For better visibility, only 8 mm of the centerline is shown, corresponding to ten extrusions in N14 and five extrusions in N19.

While the downward spikes are almost at the same level, the N19 curves spike upwards significantly higher than the N14 curves. Matching the number of extrusion lines in the observed region, the red lines of N19 peak five times, while the blue lines of N14 have more but smaller peaks. The 25% and 75% quantiles of the parameter sets confirm that the lower range is almost identical, while the upper differs significantly, with N19 having almost double the difference between them compared to N14 as listed in Table 3.

This indicates that the misaligned fibers in the extrusion line core do increase stiffness of the extrusion center compared to the edges. 

During the tests, specimens failed in the measurement region, with cracks perpendicular to the stress direction. For Y specimens, crack initiation could be seen in DIC to begin at an edge of the specimens between the extrusion lines.

## 4. Discussion

The results show a large impact of extrusion parameters on the alignment of fibers during FFF. The range in which the fiber alignment can be influenced was seen to have significant impact in the mechanical properties of printed parts. Recalling the effects of parameters shown in Figure 9, the ratios of line width and layer height over nozzle diameter, as well as their interaction, had a stronger impact than the actual extrusion line width shown by *w* × *D* in the diagram. These ratios describe the convergence or divergence of material flow when exiting the nozzle and flowing into the shape of the extrusion line. All samples had a higher fiber misalignment than the filament used for printing which had a standard deviation of 5.4°, and also higher than an extrusion into air that had a standard deviation of 7.4°, which increase was probably due to die swell effects as described in [44]. The conducted research shows that the flow at the nozzle exit reshaping the material from a Z aligned cylinder into a X aligned rectangle already increases the fiber misalignment, which is in agreement with the observations made by [48]. This indicates that, of the four phases described in the beginning of the article, phase one, the flow inside the nozzle, seems to have little effect on final fiber alignment, as the following phases reshape the material structure. Phases two and three, the convergent and divergent flow shaping the extrusion line are likely to happen at the same time, rather than sequentially and appear to be the flow condition dominating fiber alignment. Finally, from the experiments it seems like phase four, the stretching of the deposited material can have a negative impact on the extrusion line geometry by tearing the material from the extrusion line at high speeds and low layer heights resulting in high stress. 

Tested along X, a higher fiber misalignment reduced the tensile strength, however improved the strain at break, leading to similar if not slightly higher toughness. This could be an interesting option for tailoring the energy absorption rate of 3D printed parts. The fibers with high misalignment to X were found mostly in the core of the extrusion lines resulting in local differences of stiffness and strain, however the aligned fibers at the extrusion line edge prevented the fibers in the core from improving the strength in the Y direction. The fiber alignment of the initial experiments on single walled cubes could be transferred to thicker tensile test specimens. This means that fiber alignment can be controlled in parts of changing thickness by the extrusion parameters of the individual extrusion lines, without having to build the detailed simulations common in injection molding.

Additionally, the impact of parameters on surface quality was briefly analyzed. Comparing the contour plots, the research suggests that parts printed with high fiber alignment are likely to have a low surface roughness and high fiber misalignment at large layers leads to poor surface quality, as large layers introduce a surface waviness. 

The fiber orientation in this article was investigated through cross-section micrographs, which may result in different values for the misalignment of the fibers depending on the position of the cross-section in the extrusion line. Micro-CT images would help to produce more accurate results. For a high number of parameter sets and with several images per parameter set, micrographs offer a quick and inexpensive way to screen for major effects with sufficient accuracy. 

Finally, regarding model quality from the DoE used to generate the contour plots based on the effects of the previous sections needs to be addressed. Modde calculates four parameters to judge the model quality:R2 shows the model fit and should be larger than 0.5 for a significant model.Q2 shows an estimate of prediction precision and should be larger than 0.5 for a good model.Model validity tests diverse model problems and should be larger than 0.25.Reproducibility is the variation of the replicates and should be larger than 0.5.

Figure 17 shows the indicators for the two models of fiber orientation and surface roughness. All indicators are above their minimum requirements, the model for fiber orientation showing excellent values in all indicators. Reproducibility is lower for the fiber orientation than for surface roughness, which may be caused by micrograph measurements. The values still indicate a very good model.

The results presented here fit well with observations that fiber alignment in printed parts can be very high and used for functionalization of parts such as in [48]. However, extrusion parameters were found that increase fiber misalignment significantly. These parameter sets may be unusual for FFF however in large scale systems using screw extruders to process granular feedstock, higher and wider extrusion lines are more common, including extrusion lines significantly wider than the nozzle. Further investigations on this scale should yield interesting insights.

Reshaping of extrudate into extrusion lines is the most important factor for fiber orientation, indicating that focusing on the flow and fiber alignment only inside the nozzle may be disregarding critical flow conditions for fiber alignment in the part. At extrusion line widths close to the nozzle diameter the flow can be assumed to be planar, experimentally confirming previous simulation work [45,46], resulting in high fiber alignment. At layer lines significantly larger than the nozzle diameter, the material is forced to flow also in Y direction, changing from planar to 3D flow, introducing a new fiber alignment direction. The experiments of this article cannot answer whether a higher or lower fiber alignment of fibers by different nozzle design [44,49] after phase one or already present in the filament feedstock translates through phases two and three or is completely rearranged by them, as only one type of material and nozzle design was used. 

## 5. Conclusions

To conclude this article, we answer the questions posed in the introduction: Whether fiber orientation and alignment can be controlled by the extrusion parameters and if they can be influenced enough to have a significant impact on the material properties of printed parts. It has been shown that it is possible to control the fiber orientation through extrusion parameters and they impact the material properties significantly. The parameter sets selected for tensile testing differed only in extrusion line width and material flow velocity, tripling the misalignment. Thereby reducing the stiffness by more than half and strength by just over 40%, but increasing strain at break by over 75%, slightly increasing the material toughness when tested along the extrusion direction X. Perpendicular to the extrusion lines within the layers along Y, the difference in fiber alignment had less impact on the mechanical properties on a macroscopic scale. However, the strain fields showed that the local elastic moduli were affected and fiber misalignment leads to a much more inhomogeneous strain field. Macroscopically a higher fiber misalignment reduces the anisotropy between X and Y, yet this is achieved by sacrificing strength and stiffness in X without improvement along Y.

When designing a printing process to maximize the part strength it is important to consider the effect of extrusion parameters on the fiber alignment as a large nozzle with low flow divergence achieves better fiber orientation than a small nozzle with high flow divergence, as evidenced by experiments N2 and N17 that print almost identical extrusion line size but have very different fiber alignments. While the flow of N2 had to significantly expand in the Y direction, the flow of N17 is completely planar, resulting in the alignment seen in Table 4.

Flow velocity had a low impact on fiber orientation and surface roughness. Therefore, the print speed can be used to control the layer time and optimize prints towards high layer bonding strength.

## Figures and Tables

**Figure 1 polymers-13-02443-f001:**
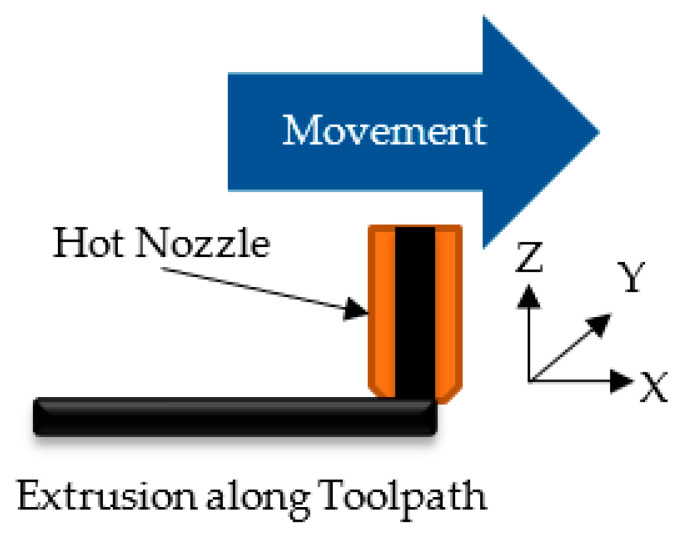
Local coordinate system used to describe the material properties.

**Figure 2 polymers-13-02443-f002:**
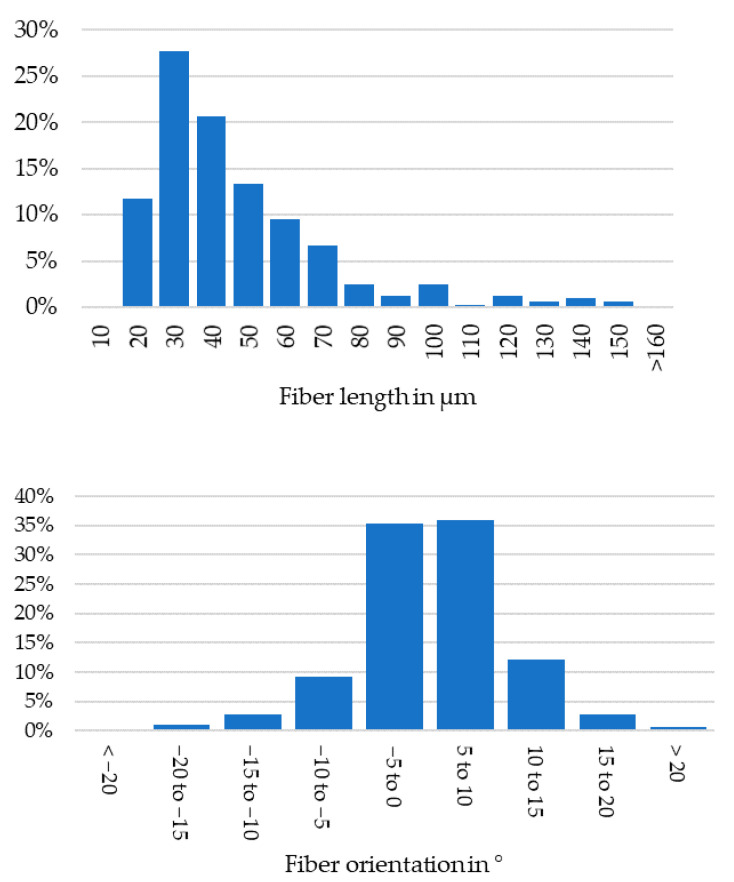
Top: Fiber length distribution in filament; Bottom: Fiber orientation in filament; Standard deviation of fiber orientation is 5.4°.

**Figure 3 polymers-13-02443-f003:**
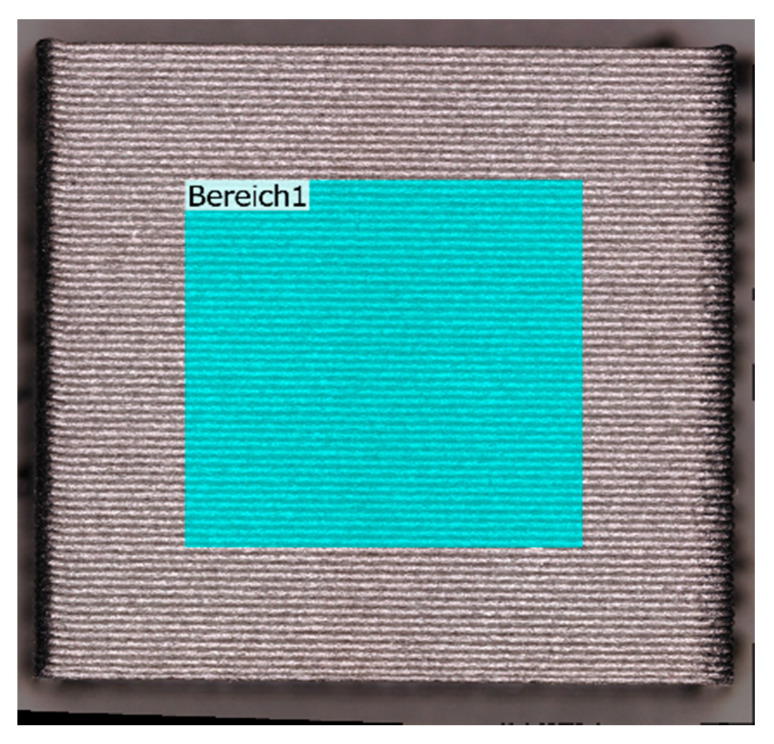
Example of surface roughness scanning area on experiment N14.

**Figure 4 polymers-13-02443-f004:**
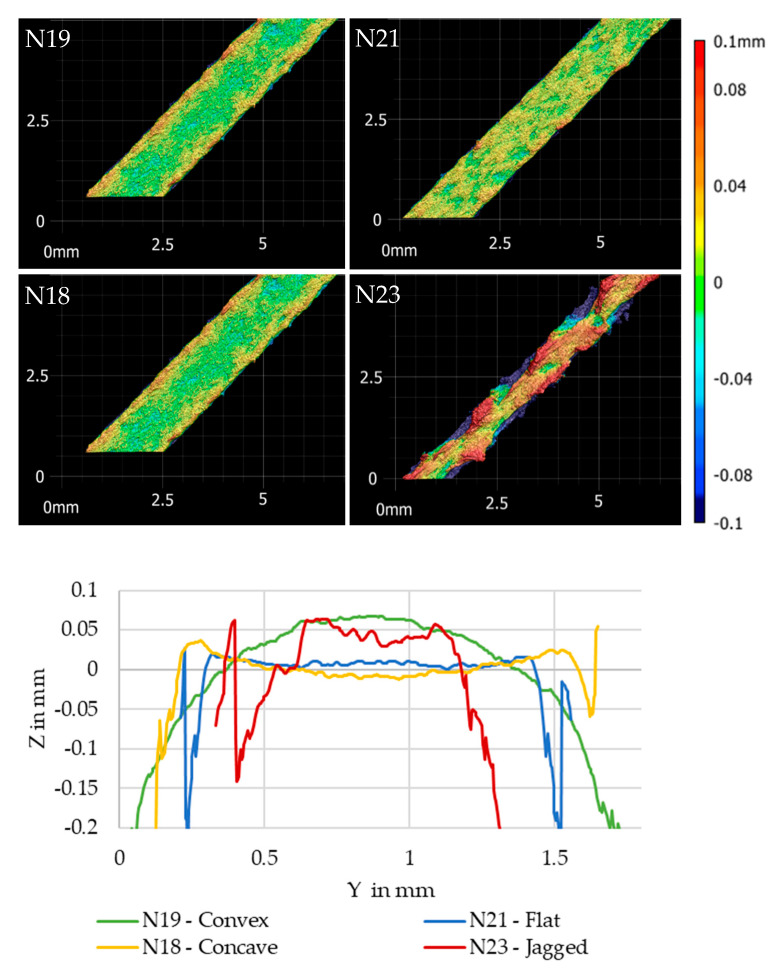
Types of extrusion line shapes and their height profiles.

**Figure 5 polymers-13-02443-f005:**
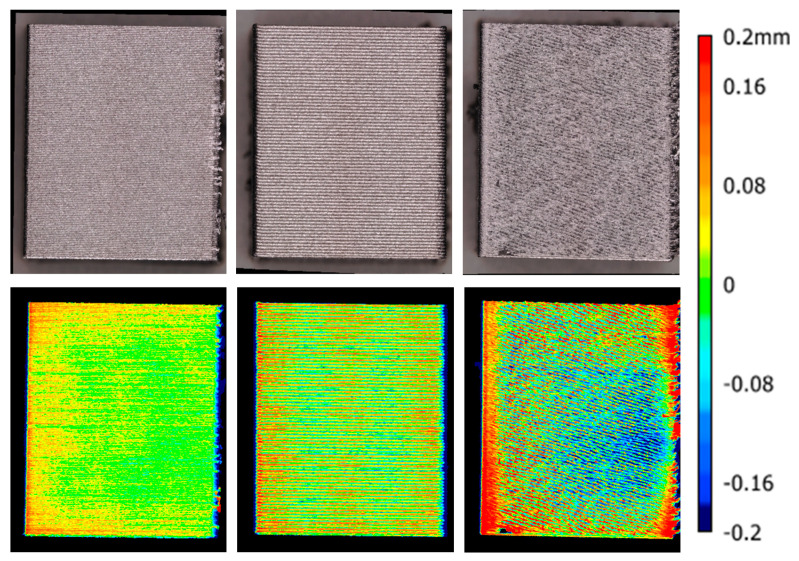
Examples of different surface qualities: Left: N9 with the lowest measured surface roughness; Center: N14 with large layer lines; Right: N23 with artefacts due to jagged extrusion lines.

**Figure 6 polymers-13-02443-f006:**
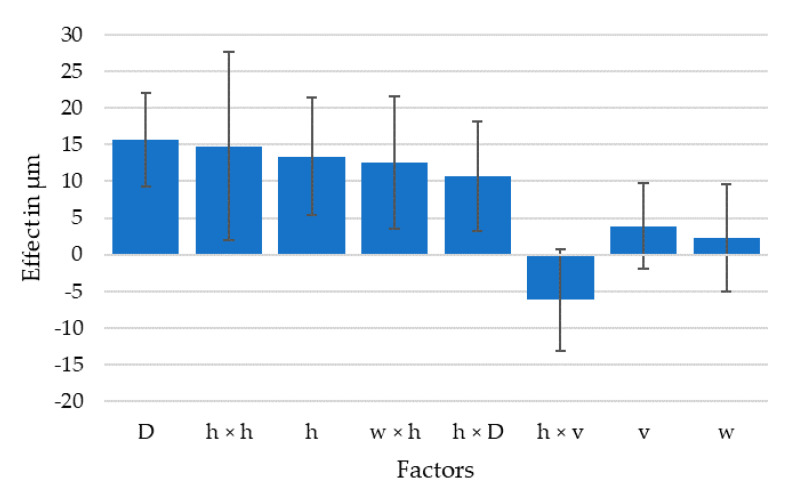
Effects of model parameters on surface roughness in descending order of magnitude for 95% confidence interval, Y axis is the effect on surface roughness if the parameter is varied between its upper and lower limit.

**Figure 7 polymers-13-02443-f007:**
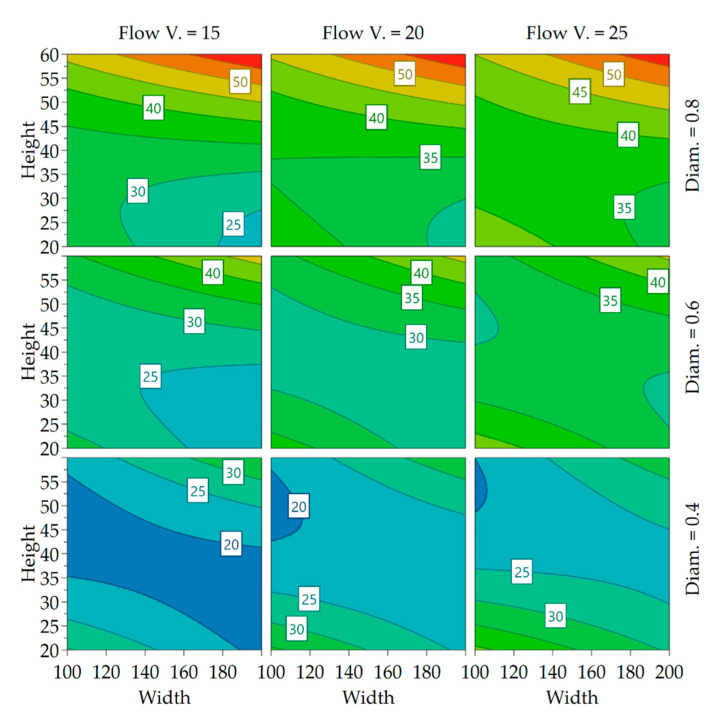
4D contour plot of surface roughness.

**Figure 8 polymers-13-02443-f008:**
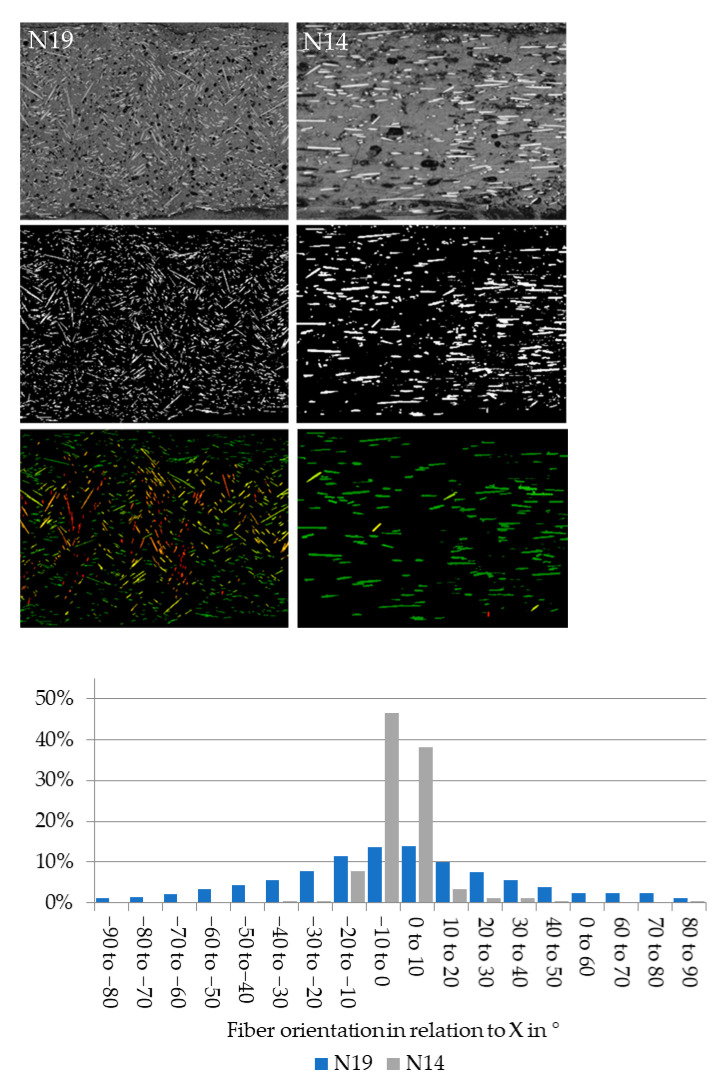
Top: Steps in measuring the fiber orientation, grayscale, binary and color-coded by orientation: Left: N19#2 with a high fiber misalignment of 34.7°, measured in 1986 detected fibers in the image; Right: N14#3 with high fiber alignment of 9.5°, measured in 181 detected fibers; Bottom: the resulting fiber orientation distributions.

**Figure 9 polymers-13-02443-f009:**
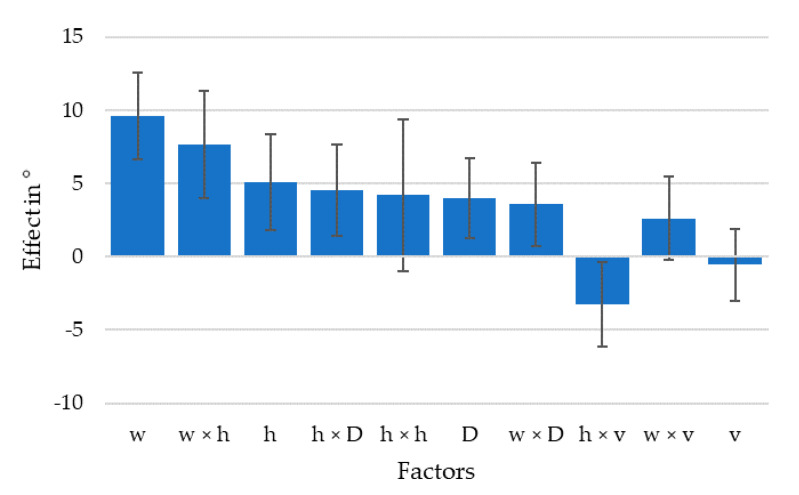
Effects of extrusion parameters on fiber alignment sorted by magnitude for 95% confidence interval, Y axis is the effect on fiber orientation if the parameter is varied between its upper and lower limit.

**Figure 10 polymers-13-02443-f010:**
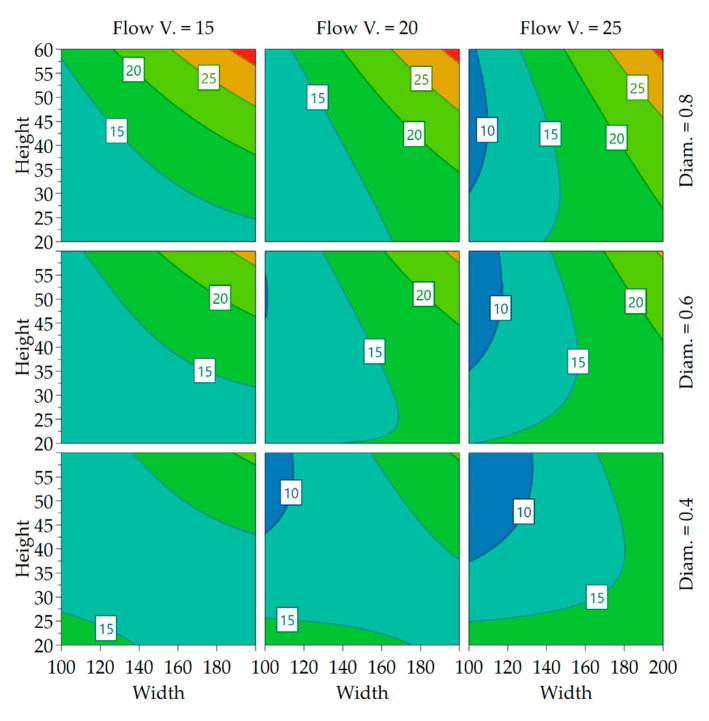
4D contour plot of fiber orientation labels showing the standard deviation of fiber alignment of the levels, Width and Height in %, Flow Velocity in mm/s and Diameter in mm.

**Figure 11 polymers-13-02443-f011:**
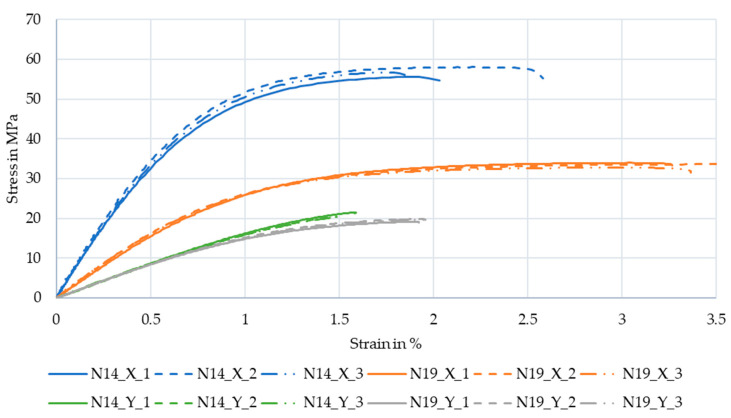
Stress-strain curves of the tensile tests; Legend entries are “experiment number”_”test direction”_”repetition number”.

**Figure 12 polymers-13-02443-f012:**
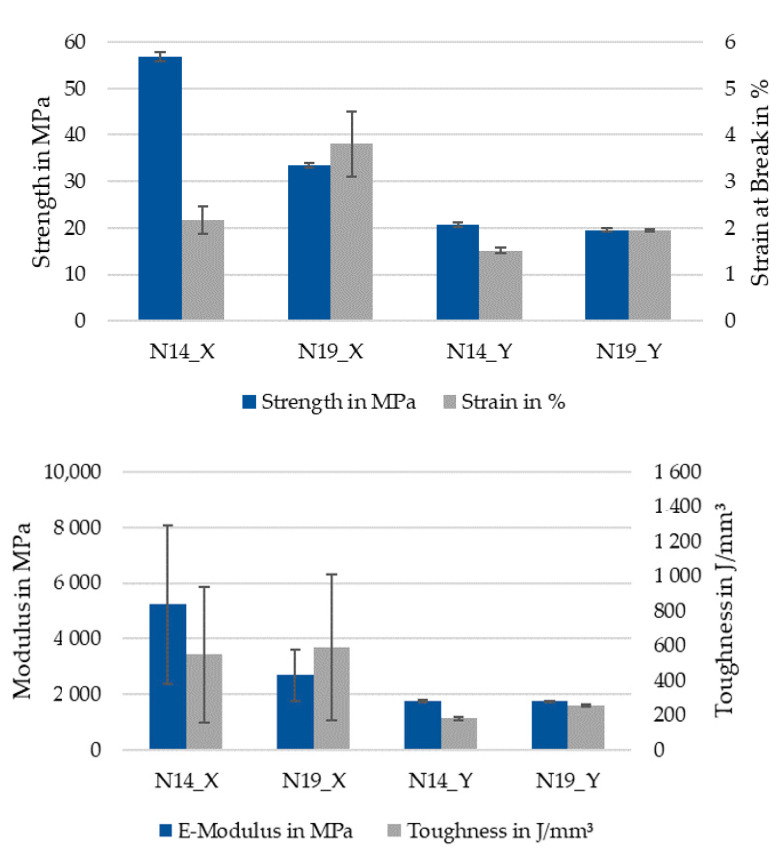
Average values of the tensile tests: Top: strength and strain at break; bottom: stiffness and toughness.

**Figure 13 polymers-13-02443-f013:**
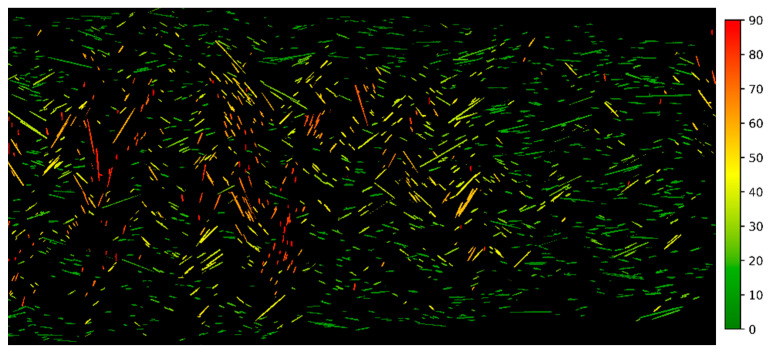
Fiber orientation in N19 with color coded deviation from X.

**Figure 14 polymers-13-02443-f014:**
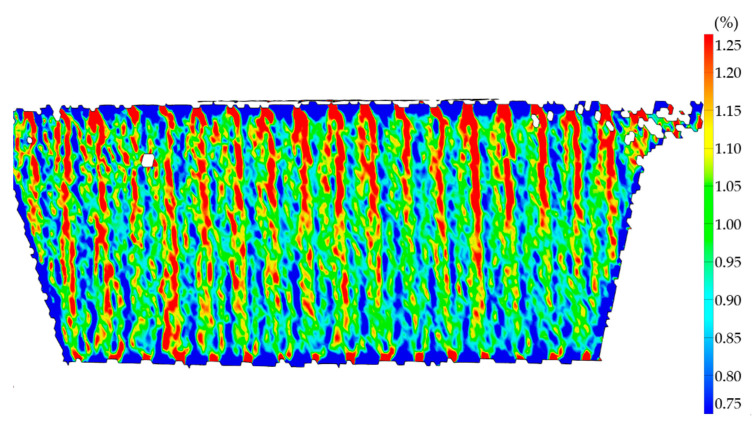
Specimen N19 tested along Y with high fiber misalignment in DIC at 1% average strain along 25 mm of center line.

**Figure 15 polymers-13-02443-f015:**
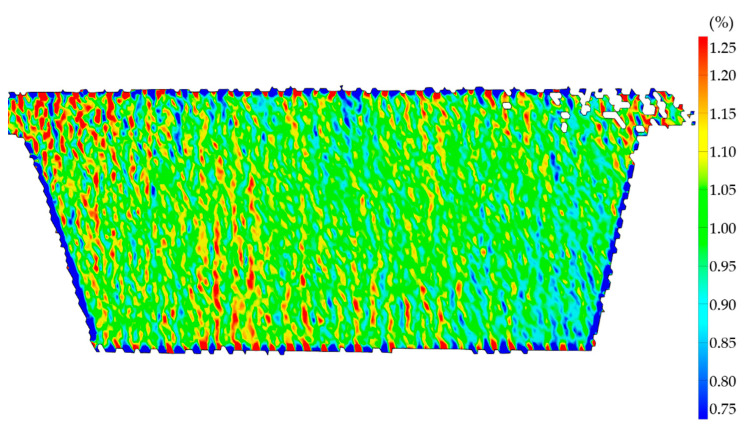
Specimen N14 tested along Y with low fiber misalignment in DIC at 1% average strain along 25 mm of center line.

**Figure 16 polymers-13-02443-f016:**
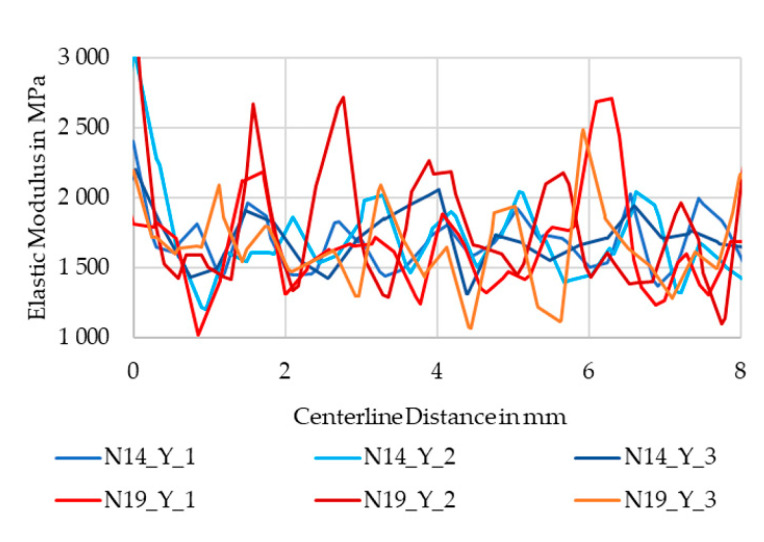
Local elastic modulus calculated between 10 MPa and 5 MPa.

**Figure 17 polymers-13-02443-f017:**
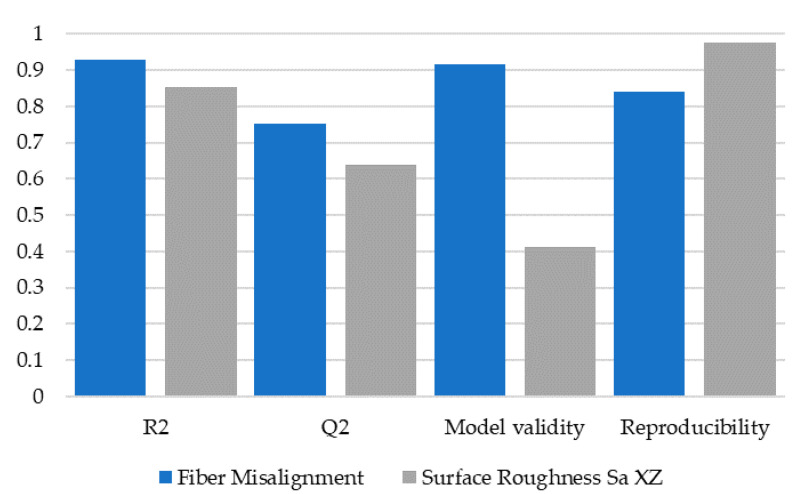
DoE model quality indicators.

**Table 1 polymers-13-02443-t001:** Investigated parameters, their limits and units.

Parameter	Lower Limit	Upper Limit	Unit
*w*: Extrusion width/nozzle diameter	100	200	%
*h*: Layer height/nozzle diameter	20	60	%
*D*: Nozzle diameter	0.4	0.8	mm
*v*: Flow velocity at nozzle outlet	15	25	mm/s

**Table 2 polymers-13-02443-t002:** Averaged values for strain at break, strength, stiffness and toughness of the specimens.

	Strain in %	Strength in MPa	E-Modulus in MPa	Toughness in J/mm^3^
N14_X	2.17	56.80	7328.39	953.85
N19_X	3.81	33.47	3375.88	1051.29
N14_Y	1.51	20.74	1759.86	180.68
N19_Y	1.95	19.58	1750.65	253.66

**Table 3 polymers-13-02443-t003:** 25% and 75% quantiles and difference of local modulus curves of N14 and N19.

Sample Group	Modulus Q25 (MPa)	Modulus Q75 (MPa)	Difference (MPa)
N14	1523.63	1813.78	290.15
N19	1514.37	2045.30	530.93

**Table 4 polymers-13-02443-t004:** Comparison of N2 and N17 extrusion line geometry and fiber alignment.

Experiment	Extrusion Line width (mm)	Extrusion Line Height (mm)	Flow Velocity (mm/s)	Nozzle Diameter (mm)	Fiber Misalignment (°)
N2	0.8	0.24	15	0.4	22.13
N17	0.8	0.264	25	0.8	9.97

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
