# Peer review of "Effect of Extrusion Parameters on Short Fiber Alignment in Fused Filament Fabrication"

_polymers, 2021, doi:10.3390/polym13152443_

Round 1

Reviewer 1 Report

Dear Authors,

in your interesting manuscript, the following points should be added/changed to further improve it:

  • Abstract: What is D-optimal?
  • I am not sure about your use of the word "bead". Actually a bead is something droplet-shaped, but here you use it for the whole strings placed on the printing bed or the previous layers. I know that it's technically also used for a whole metal welding line or the like, but from a polymer point of view it sounds unusual for me.  Please check whether "bead" can be used here.
  • line 59: Please define DoE (and there shouldn't be an apostroph).
  • line 119: line width
  • Table 1: The flow velocity is not independent from the other parameters. Why can you nevertheless chose it as a parameter (instead of the better measurable printing speed)?
  • lines 144 ff: Please explain this part a little bit more in detail. Not all polymer or 3D printing specialists will know all special version of DoE. Talking about 30 experiments, how many specimens were printed per experiment?
  • 2.2.2: Which resolution / accuracy does the surface roughness measurement have? Giving the name of an experiment in Fig. 2 is useless as long as the experiments are not defined.
  • lines 178 ff: Please explain these fiber orientation measurements more in detail. Howe were they performed, with which instruments etc.?
  • What is completely missing in Section 2 are the simple basic printing parameters - temperatures, sample dimensions, perimeters and filled top / bottom layers, infill pattern and orientation, etc. And, naturally, the aforementioned list of samples.
  • Okay, in Section 3 there is a hint to Table 5 (this hint must be given in Section 2). So next questions about this table: What is meant with "extrusion bead shape"? Why are there less samples with 0.6 mm nozzle width? How was the flow velocity measured? It would definitely make more sense to have the experimental settings first (actually better in Section 2), followed by the measured results in Section 3. What is meant with "Real width of set width"? Can all the measured and set values really be given with up to 5 significant digits? Please add the standard deviations to see how exact these values are. And units must not be given in square brackets.
  • Fig. 4: What you show here is actually a waviness, not a roughness.
  • Fig. 6: It doesn't make sence to put units separately for single values, it is even shorter to write "flow velocity = 15 mm/s" etc.
  • Fig. 7: The numbers in the histograms are not readable. Howe many fibers were used to prepare them?
  • line 309: "Entering the standard deviations ..." - in which program?
  • Fig. 8: The axis labels are too small (in Fig. 5 actually also, but this is shown in a larger scale).
  • line 341: What is special about N14 and N19? And what about the other samples mentioned in this paragraph?
  • Table 2: Again, please add standard deviations and fit the numbers of significant digits correspondingly. And please change from decimal commas to decimal dots (not only here).
  • Fig. 11: The lower parts of the error bars on the blue bars are not visible.
  • line 376: Standard deviations are given with max. 2 digits, so it should be 30° (ditto in the residual text).
  • Fig. 12: How was coloring of the fibers performed, by ImageJ?
  • Fig. 15: MPa on the y-axis.
  • Table 4: Is there anything special about these two samples, making them worth being shown in the conclusion, such as best and worst fiber misalignment? Besides, is it fair to compare bead width of 100 % and 200 % of the nozzle diameter?
  • References: Most of them are relatively old - please check whether there is no more recent literature dealing with the influence of different printing parameters on mechanical properties, surface roughness etc.

Author Response

Dear Reviewer,

thank you for taking the time to review our article and the detailed comments. They were very clear and helpful.

Overall major revisions were necessary which resulted in a lot of changes to the original document. The main critique points were on the introduction with the background which was not detailed enough and the references were quite old and the presentation of the methodology and results which was difficult to read.

I believe the changes made fix this. The introduction was extended to consider more background of newer sources almost doubling the number of references with sources mostly newer than 2017. The methodology was changed with several clarifications based on the comments made by all reviewers, all figures were redone and formatted to the same template.

To answer your comments specifically:

  • The D-Optimal DoE type was taken out of the abstract and explained in the text in section “2.2.1 Parameter Identification & Design of Experiments” line 244ff.
  • I also added an explanation why we chose flow velocity and how we define it, the short reason is that it makes sure the extruder is able to provide the flow, and that the flow under the nozzle is more comparable. With layer widths from 0,4 to 1,6 mm and layer heights from 0,08 to 0,48 mm a fixed print speed results in huge differences of material flow rate making it very difficult to set the limits correctly. Calculating print speed from flow rate makes this a lot easier.
  • I added the simplify3D parameters used for toolpath generation as another table in the appendix.
  • Fiber orientation measurement is done by python with a code we wrote, basically it identifies objects, calculates their eigenvectors and eigenvalues and from this then the length and orientation. I’d be happy to provide the python code, however all comments in it are in German and in the short time we had for the revisions it was unfeasible to translate and revise the article.
  • As suggested I added the hint to table 5 earlier in the text.
  • Also I went into detail about the surface waviness and superimposed surface roughness caused by the sharp surface artefacts of jagged extrusion lines.
  • I reformatted all graphs, figures and histograms to make them uniform and readable.
  • Coloring of the fibers is done in python based on their calculated orientation using the following code:
    • new_cmap = matplotlib.colors.LinearSegmentedColormap.from_list(
    • "", ["green", "yellow", "red"])
    • # Set color for background
    • set_bad(color='black')
    • # limits for the colors
    • vmin = 0
    • vmax = 90
    •  
  • The image is then output with the colors
    • # Insert image, colored by colormap 'new_cmap'
    • imshow(delta_labeled, cmap=new_cmap)
  • Regarding the references I made sure to search for references newer than 2017 when extending the background of the introduction.

Thank you again for taking the time to review our article. I hope with these revisions made you can recommend the article for publication.

Best regards,

Patrick Consul

Reviewer 2 Report

- The fiber alignment resulting from the suitably designed toolpath is mainly related to the rheological properties of the processed material. In my opinion, authors must comment and supplement the revised manuscript version. The correlation between the viscosity and flow behavior description with technical parameters in the manuscript is insufficient. The fiber orientation depends to the greatest extent on the flow and rheological properties of the composite thermoplastic materials, and omission of this fact is quite a failure.

- Were the materials dried before being shaped? If so, please complete the information on the initial preparation of the composition. Unfortunately, the authors also omitted a broader description of the research materials used. Please, present the broader filament characteristics in the materials section. It is not sufficient to simply display the trade names and the fiber content.

- The quality of the charts should be improved. This applies to both their form and the way of presentation. Graphs made, each in a different style in Excel, should be standardized where possible.

- Presentation of the obtained measurement data and methodology is quite chaotic and unreadable. I recommend systematizing the presented results. The presented conclusions should be reformatted, the table with data should be included in a separate section in the discussion part of the manuscript.

- Unfortunately, the submitted work presents a fairly extensive but typical 3D printing articles case study without in-depth analysis. The manuscript should be edited to translate the obtained research results into technical recommendations for other materials than those examined by the authors. Moreover, the authors emphasize that the results are not entirely clear (Lines 481-487). Considering that the authors throughout the work refer to "material flow" without any reference to rheological properties, but only printing parameters, the presented work has the features of a research report rather than serious scientific work. Consequently, submitted work should be significantly modified if it is to be recommended for publication.

- Present extension by DoE as the first one used.

Author Response

Dear Reviewer,

thank you for taking the time to review our article and the detailed comments. At first I was a little put off as they are quite hard but as I read them with the text I must agree that they are correct. The first version was a bit rushed and the clarity and especially the figures quality suffered as a result, thank you for reading all the way through regardless of this.

Overall major revisions were necessary which resulted in a lot of changes to the original document. The main critique points were on the introduction with the background which was not detailed enough and the references were quite old and the presentation of the methodology and results which was difficult to read.

I believe the changes made fix this. The introduction was extended to consider more background of newer sources almost doubling the number of references with sources mostly newer than 2017. The methodology was changed with several clarifications based on the comments made by all reviewers, all figures were redone and formatted to the same template.

To answer your comments specifically:

  • Viscosity and flow behavior are very closely related as you pointed out, and I did test with different materials in a screening before these experiments using two other polymers, and fiber contents. While the fibers were not exactly the same, the trend was. Out of plane flow for extrusion lines wider than the nozzle always led to fiber misalignment and the differences were actually quite small. We also checked with different temperatures and the impact was small as well. Extrusion temperature is the main driver of layer bonding which is the most critical aspect in extrusion based AM so it is not really a factor that can be changed. The same way the material is often fixed by the application. So I decided to leave these parameters out of the study because their impact was relatively low in preliminary screening and the control we have over them in process design is also low. I did go into a lot more detail in the introduction to explain why this is more critical in injection moulding and is studied in much more detail. The main result of this article is that fiber orientation in the part is determined by the individual extrusion lines which makes it much easier to predict.
  • The materials were dried indeed, I had it in one of the first drafts but must have deleted it during internal review by accident. I also added more details on the material, including the fiber length and orientation in the feedstock which is probably the most critical.
  • Charts were awful, they are better now. Sorry about that.
  • I added explanations to the methodology to make it clearer, a big issue is that the 3D scans which were intended as quality control have quite interesting results themselves which distracts from the main intention of the article. I added explanations that those sections are to ensure the process is stable but not the main objective of the article. I think it is clearer now. I left table 5 in the Appendix though as it disrupts the reading flow if it is in the text.
  • 3D printing works are indeed often very extensive and relatively superficial, but as the technology is relatively young compared to other polymer processing technologies this should be understandable. There is much more that has not yet been studied in detail in 3D printing and even superficial results are necessary to show the areas in which future research needs to go into detail. An example is the out of plane flow that misaligns the fibers, that is not covered by Ref45 and Ref46, which both go into detail about material flow and rheology but neglect the out of plane flow. I think the lines you quote about the results not being clear, may have been misunderstood. In no way are we saying the results are not entirely clear, we are describing the quality markers of the DoE, a lower reproducibility of 0,84 for fiber orientation to 0,98 for surface roughness is still a very good reproducibility. In fact indicators all point to a very good fit of the model.
  • We were unable to understand the last comment so I could not consider it in the revisions, I hope this was something minor.

Thank you again for taking the time to review our article. I hope with these revisions made you can recommend the article for publication.

Best regards,

Patrick Consul

Reviewer 3 Report

Comments

This paper studied the effect of extrusion parameters on short fiber alignment. The outcome is interesting for readers. However, there are several aspects that need to be improved. The reviewer can only recommend for publication if the author satisfactorily address the following major comments in the revised version.

  1. The justification of selecting parameters in Table 1 should be discussed. Why these particular parameters were selected for investigation?
  2. The colure of the legend in Fig 10 is difficult to differentiate. For example, the first three colours seems very similar. Need to revise this figure.
  3. The failure mechanism of the specimen should be discussed more clearly.
  4. How many replicate samples were tested in each category?
  5. The novelty of the study should be highlighted more clearly at the end of introduction section. How this study is different from the published study in literature?
  6. How the outcome of this study will benefit researchers and end users? This need to be highlighted in introduction or end of conclusion.
  7. The recent investigation on the application of fibre composites should be discussed in introduction section to improve the background study. Recently, fibre composites are implemented in many structural application such as railway sleepers [Ref: Static behaviour of glass fibre reinforced novel composite sleepers for mainline railway track], fire resistant structures [Ref: Effect of fire-retardant ceram powder on the properties of phenolic-based GFRP composites] and structural repair system [Ref: State-of-the-art of prefabricated FRP composite jackets for structural repair]. Suggest to include them in introduction section with proper citations to improve the background study.

I would be happy to see the revised version to understand how these comments are being addressed.

Author Response

Dear Reviewer,

thank you for taking the time to review our article and the detailed comments. They were very clear and helpful. I also read the references you suggested and added them in the introduction, thank you for this. I also found them very interesting to read as they are from a very different field of application than our usual.

Overall major revisions were necessary which resulted in a lot of changes to the original document. The main critique points were on the introduction with the background which was not detailed enough and the references were quite old and the presentation of the methodology and results which was difficult to read.

I believe the changes made fix this. The introduction was extended to consider more background of newer sources almost doubling the number of references with sources mostly newer than 2017. The methodology was changed with several clarifications based on the comments made by all reviewers, all figures were redone and formatted to the same template.

To answer your comments specifically:

  1. I added some more detail on why these were selected, also I added references I had previously missed that did 2D simulations on the flow which found these to be the most relevant as well, however the left out extrusion width as this dimension was not modeled by their planar flow model.
  2. I made all figures again and reformatted them to a uniform template, the first version of images, was very rushed regarding this to meet the submission deadline. Sorry about those .
  3. I added a sentence about it in line 590ff. In general the failure mechanism was as you would expect it to be, within the measurement are and perpendicular to the stress. For Y specimens it was visible during DIC that the crack started between the extrusion lines at the edge of the sample. This is the usual way for 3D printed specimens to fail, so I did not go into to much detail.
  4. I added this information in the methodology, in short summary: 1 Cube was printed for each experiment, of this cube 3 different locations were checked for the fiber orientation which was then averaged for the DoE input, for tensile testing it was 3 specimens for each parameter set, had very little scattering of the curves as shown in the stress strain curve of figure 11.
  5. I went into more detail about it in the introduction. Short summary, recent publications are mostly simulations and assume planar flow, this article considers the impact of out of plane flow in wide extrusion lines to show a possibility to change the fiber orientation and material behavior.
  6. Closely related to the previous, the benefit to researchers and users is the knowledge that fiber orientation inside a part can be controlled by the flow under the nozzle and is then fixed. This way the fiber orientation in parts can be ensured without the extensive simulations necessary for injection moulded parts. Additionally it was shown that while stiffness and strength are reduced with higher fiber misallignment, the higher strain at break results in similar material toughness, opening up interesting possibilities to tailor the energy absortption rates of parts that could find application in crash relevant elements.
  7. As mentioned above I expanded the background significantly including your three suggestions. Thank you for pointing me to them, I was previously not aware of them.

Thank you again for taking the time to review our article. I hope with these revisions made you can recommend the article for publication.

Best regards,

Patrick Consul

Round 2

Reviewer 2 Report

The authors corrected the text as recommended and responded to the comments presented in the review, the manuscript may be recommended for publication.

The last comment concerned the DoE abbreviation extension in the text when it was first used.

Reviewer 3 Report

I have no further comments